# Long Non-Coding RNAs in Liver Cancer and Nonalcoholic Steatohepatitis

**DOI:** 10.3390/ncrna6030034

**Published:** 2020-08-29

**Authors:** Shizuka Uchida, Sakari Kauppinen

**Affiliations:** 1Cardiovascular Innovation Institute, University of Louisville School of Medicine, Louisville, KY 40202, USA; 2Center for RNA Medicine, Aalborg University, DK-2450 Copenhagen SV, Denmark

**Keywords:** liver, lncRNA, miRNA, liver cancer, HCC, NASH

## Abstract

This review aims to highlight the recent findings of long non-coding RNAs (lncRNAs) in liver disease. In particular, we focus on the functions of lncRNAs in hepatocellular carcinoma (HCC) and non-alcoholic steatohepatitis (NASH). We summarize the current research trend in lncRNAs and their potential as biomarkers and therapeutic targets for the treatment of HCC and NASH.

## 1. Introduction

Liver cancer is the second leading cause of cancer-related death globally with an estimated number of one million cases diagnosed in 2016 and a 5-year survival rate of less than 20% [1,2,3]. Hepatocellular carcinoma (HCC) accounts for 90% of primary liver cancers and is increasing in incidence worldwide. Major risk factors for HCC include chronic hepatitis B virus (HBV) and hepatitis C virus (HCV) infections [1,4], alcohol-induced liver disease, non-alcoholic fatty liver disease (NAFLD), which was recently proposed to be renamed as metabolic-associated fatty liver disease (MAFLD) [5] and non-alcoholic steatohepatitis (NASH) [6]. HCC development is recognized as a multistep process with 70–80% of cases occurring in the context of liver cirrhosis, in which multiple somatic, genomic, and epigenomic alterations occur along each stage of disease progression [1]. HCCs can be classified by genomic, epigenomic, and transcriptomic profiling into two major molecular subtypes, each comprising ca. 50% of HCC patients. The first class, named as the proliferation class, is associated with poor prognosis, chromosomal instability, and activation of classical oncogenic signaling pathways involved in cell proliferation and survival. The non-proliferation class of HCC is more heterogenous, is less aggressive with slower disease progression, and is associated with better prognosis than the proliferation class tumors [1].

HCC is highly resistant to therapy and thus considered as a difficult-to-treat cancer. For HCC patients with early-stage disease, potentially curative treatment options (e.g., liver transplantation, local ablation or surgical resection) are possible [1,4]. However, most HCC patients are diagnosed at an advanced tumor stage and do not qualify for these treatment options. The multi-target tyrosine kinase inhibitor (TKI) sorafenib is the first systemic therapy approved for the treatment of advanced stage HCC and has been shown to extend the median overall survival from 8 to 11 months [1,4]. Improvements in HCC patient outcome has also been reported with other TKIs (i.e., lenvatinib, cabozantinib, and regorafenib) as well as with nivolumab, which is a monoclonal antibody targeting the immune checkpoint PD-1 [1]. However, the median overall survival of advanced stage HCC patients is still only 11–12 months, which underscores the unmet medical need for novel therapeutic approaches for treatment of HCC.

NASH is a progressive subtype of NAFLD (MAFLD) characterized by the presence of ballooned hepatocytes, inflammatory infiltrates, and fibrosis [7,8]. It is a complex and multifactorial disease that is modulated by several mechanisms including genetic, environmental, metabolic, and gut microbial factors [7,8]. Genome-wide association studies (GWAS) in large cohorts indicate that polymorphisms in two genes, patatin-like phospholipase domain–containing 3 (*PNPLA3*) and transmembrane 6 superfamily member 2 (*TM6SF2*), are strongly associated with development of NAFLD and NASH [7,8]. Furthermore, gut microbiota has been shown to impact host susceptibility to obesity, hepatic steatosis, NASH, and liver fibrosis, whereas the most important environmental factors linked to NASH include dietary habits, activity, and socioeconomic factors [7,8].

NASH is strongly associated with obesity and the metabolic syndrome and affects over 6% of the adult population in the United States. Furthermore, 15–25% of NASH patients develop liver-related complications, such as cirrhosis and HCC [7,8,9]. Indeed, NASH is rapidly becoming the leading cause of end-stage liver disease and liver transplantation [10]. The annual medical costs associated with NAFLD exceed USD 100 billion in the United States and EUR 35 billion in four European countries (Germany, France, Italy and the UK); much of which is attributable to NASH [9]. This underscores the importance of developing new effective and safe therapies for treatment of NASH.

Recent data imply that the human genome is pervasively transcribed and encodes tens of thousands of long noncoding RNAs (lncRNAs) that play regulatory roles in numerous biological processes [11,12,13]. LncRNAs comprise a large and heterogeneous group of transcripts that are longer than 200 nucleotides (nt) in length, lack an open reading frame, are often expressed at much lower levels than protein-coding genes, and are less conserved than mRNAs [11,12,14,15]. Many lncRNAs exhibit highly cell- and/or tissue-specific expression and play important roles as regulators of key cellular pathways, which makes them potential targets for therapeutic intervention [16,17]. In this review, we summarize the current understanding of lncRNA functions in the liver. Furthermore, we review the recent progress on the role of lncRNAs in the pathogenesis of HCC and NASH and discuss their potential as biomarkers and targets for therapeutic intervention.

## 2. Molecular Mechanisms of Long Non-Coding RNAs in the Liver

### 2.1. LncRNAs as Scaffolds for Proteins

To date, several mechanisms of action for lncRNAs have been reported. The most investigated one is a molecular mechanism in which lncRNAs function as scaffolds for epigenetic factors, especially for the functional enzymatic component of the polycomb repressive complex 2 (PRC2), the enhancer of zeste homolog 2 (EZH2). There are few studies in recent years reporting lncRNAs binding to EZH2 in the liver, such as cancer susceptibility 11 (*CASC11*) [18], DDX11 antisense RNA 1 (*DDX11-AS1*) [19], deleted in lymphocytic leukemia 2 (*DLEU2*) [20,21], HOXA11 antisense RNA (*HOXA11-AS*) [22], nuclear paraspeckle assembly transcript 1 (*NEAT1*) [23], and Pvt1 oncogene (*PVT1*) [24] (Figure 1A). The concept of scaffolds for epigenetic factors has been heavily investigated as chromatin immunoprecipitation (ChIP) can be performed with ease using kits readily available from commercial vendors. Although it is popular to identify lncRNA binding to EZH2, it was shown to bind any RNA promiscuously [25,26,27,28,29]. Thus, such mechanism must be investigated carefully.

Besides EZH2, other epigenetic and transcription factors have been identified to bind lncRNAs to regulate transcription such as: damage-induced long noncoding RNA (*DINO*), which binds to p53 to enhance p53 stability in response to cellular stress [30]; and HAND2 antisense RNA 1 (*HAND2-AS1*), which binds to INO80 complex ATPase subunit (INO80) to induce the expression of bone morphogenetic protein receptor type 1A (BMPR1A), leading to the activation of BMP signaling [31]. Furthermore, gankyrin-associated lincRNA in hepatocellular carcinoma (*Linc-GALH*) binds to DNA methyltransferase 1 (DNMT1) to regulate its ubiquitin status [32], while the long intergenic non-protein coding RNA 324 (*LINC00324*) was shown to bind Spi-1 proto-oncogene (SPI1, also known as PU.1) to regulate the expression of Fas ligand (*FASLG*) [33]. The lncRNA regulator of Akt signaling associated with HCC and RCC (*LNCARSR*) binds to the Yes1-associated transcriptional regulator (YAP1) to inhibit its phosphorylation nuclear translocation, resulting in activation of the IRS2/AKT pathway, leading to increased lipid accumulation, cell proliferation, invasion and cell cycle [34]. The translation regulatory long non-coding RNA 1 (*TRERNA1*) binds the euchromatic histone lysine methyltransferase 2 (EHMT2) to dimethylate the promoter of cadherin 1 (CDH1) [35]; TSPOAP1, SUPT4H1 and RNF43 antisense RNA 1 (*TSPOAP1-AS1*, also known as *BZRAP1-AS1*) binds to DNA methyltransferase 3 beta (DNMT3B) to induce methylation of the promoter of thrombospondin 1 (THBS1), resulting in its inhibition [36] (Figure 1B). Taken together, these studies highlight the fact that transcriptional regulation via lncRNAs is an important mechanism to be investigated, although careful dissection of the gene regulatory networks by combining ChIP-seq and other methods (e.g., chromosome conformation capture techniques [37]) must be conducted in a genome-wide manner to record the influence of such lncRNAs on a number of genes, as many of these epigenetic and transcription factors (positively or negatively) regulate not only one but many genes.

As lncRNAs can be localized in the nucleus as well as in the cytosol, an increasing number of studies have reported lncRNAs that bind to RNA-binding proteins (RBPs). As the name indicates, the main function of RBPs is to bind single- or double-stranded RNA to regulate its metabolism, stability, subcellular localization, and translation. Such lncRNAs are often referred to as RBP sponges. In the liver, the following lncRNAs have been reported to bind RBPs: FAM83A antisense RNA 1 (*FAM83A-AS1*), which binds to NOP58 ribonucleoprotein (NOP58) to enhance the mRNA stability of family with sequence similarity 83 member A (*FAM83A*) [39]; and the gastric cancer metastasis-associated long noncoding RNA (*GMAN*), which binds to eukaryotic translation initiation factor 4B (EIF4B) to promote its phosphorylation, resulting in increased mRNA translation of anti-apoptotic proteins [40]. Furthermore, *Linc-SCRG1* (*XLOC_004166*; *lnc-Hand2-2:1*) was shown to bind the ZFP36 ring finger protein (ZFP36, also known as tristetraprolin (TTP)) to degrade its mRNA [41], whereas the long intergenic non-protein coding RNA 1093 (*LINC01093*) binds to insulin like growth factor 2 mRNA binding protein 1 (IGF2BP1) to regulate the translation of oncogene homolog 1 (*GLI1*) mRNA [42], and the LncRNA pleiotrophin downstream transcript (*lncRNA Ptn-dt*) binds to ELAV (embryonic lethal, abnormal vision)-like 1 (Hu antigen R) (Elavl1, also known as HuR) to repress the expression of *miR-96*, resulting in increased expression of anaplastic lymphoma kinase (*Alk*) [43]. Although it is attractive to investigate translation control via lncRNAs functioning as RBP sponges, just as epigenetic and transcription factors, RBPs bind many mRNAs. Thus, genome-wide screening of target mRNAs deploying RNA immunoprecipitation followed by next-generation sequencing (RIP-seq) must be conducted to understand the role of RBP sponges upon manipulation (e.g., gain/loss-of-function) of such lncRNAs.

### 2.2. LncRNAs as microRNA Sponges

Recent studies have increasingly focused on lncRNAs as microRNA (miRNA) sponges (also referred to as, competing endogenous RNAs (ceRNAs)), in which lncRNAs sequester miRNA activity (Figure 2A). The interest in studying miRNA sponges is in line with the therapeutic potential of miRNAs, since lncRNAs could be used as an additional layer of post-transcriptional control of gene expression [44]. Furthermore, over two decades of accumulated research on miRNAs has generated several experimental approaches (e.g., antimiRs, mimics, luciferase reporter assays for binding of miRNAs) that can be utilized to further elucidate the functions of lncRNAs. Since 2019, 71 lncRNAs were reported to function as miRNA sponges (Appendix A). Because the seed sequences of miRNAs are very short (6~8-mers), it is not surprising that lncRNAs contain predicted target sites for many miRNAs. For example, the lncRNA colorectal neoplasia differentially expressed (CRNDE) was suggested to sequester miR-126-5p, miR-33a, and miR-203, which target BCL2 like 2 (BCL2L2) [45], high mobility group AT-hook 2 (HMGA2) [46], and branched chain amino acid transaminase 1 (BCAT1) [47] mRNAs, respectively (Figure 2B). Furthermore, the longer a given lncRNA is, the more likely it is to contain binding sites for miRNAs [48]. For example, maternally expressed 3 (MEG3) is a maternally expressed, imprinted lncRNA with 49 isoforms (Ensemble database accession ENSG00000214548) with the longest isoform being 12,998 nt (ENST00000522771). MEG3 is a well-known miRNA sponge [49]. Since 2019, there have been four studies reporting its function as a miRNA sponge in the liver by binding to miR-9-5p, miR-21, miR-26b-5p, and miR-214, which target SRY-box transcription factor 11 (SOX11) [50], LDL receptor related protein 6 (LRP6) [51], platelet derived growth factor receptor beta (PDGFRB) [52], and activating transcription factor 4 (ATF4) [53] mRNAs, respectively. Another well-known miRNA sponge, the nuclear paraspeckle assembly transcript 1 (NEAT1) [54] with nine isoforms (ENSG00000245532) with the longest isoform being 22,743 nt (ENST00000501122), binds to let-7a, miR-22-3p, miR-29b, miR-140, miR-146a-5p, miR-155, miR-335, and miR-506 [55,56,57,58,59,60,61,62]. With an lncRNA having many isoforms, the research questions should focus on the identification of isoform-specific function of miRNA sponges, as it is experimentally very challenging to overexpress an isoform with over 10,000 nt in a stable manner. It should be noted that the ceRNA hypothesis is still being debated due to the fact that most experimental evidence is based on expression profiling such as qRT-PCR. Furthermore, studies that have modelled transcriptome-wide miRNA target site abundance suggest that physiological changes in expression levels of most individual transcripts, including lncRNAs, are insufficient to modulate miRNA activity [63,64,65,66]. Thus, further research is needed to establish the miRNA sponge (ceRNA) mechanism for lncRNAs.

One miRNA can target many mRNAs [67], and may also modulate multiple lncRNAs. Indeed, miR-33a-5p, which targets the twist family bHLH transcription factor 1 (TWIST1) and high mobility group AT-hook 2 (HMGA2), can be sequestered by two lncRNAs, namely, cancer susceptibility 15 (CASC15) [68] and eosinophil granule ontogeny transcript (EGOT) [69], respectively (Figure 2C). Another example is the lncRNA brain cytoplasmic RNA 1 (BCYRN1) and the lncRNA small nucleolar RNA host gene 15 (SNHG15), which bind to miR-490-3p targeting POU class 3 homeobox 2 (POU3F2) [70] and histone deacetylase 2 (HDAC2) [71], respectively. The reverse is also possible; that is, one gene can be targeted by multiple miRNAs [72]. Thus, it is not surprising that an lncRNA functioning as a miRNA sponge can bind several miRNAs, which target the same gene. For example, the lncRNA growth arrest specific 5 (GAS5) binds miR-21 [73], miR-23a [74], miR-135b [75], and miR-544 [76], where both miR-21 and miR-23a target the 3′-untranslated region (3′-UTR) of phosphatase and tensin homolog (PTEN) [73,74] (Figure 2D). Another example is the signal transducer and activator of transcription 3 (STAT3), which is targeted by miR-15a-3p, miR-384, and miR-4500, that, in turn, are sequestered by the lncRNA HOXA11 antisense RNA (HOXA11-AS) [77], CDKN2B antisense RNA 1 (ANRIL) [78], and small nucleolar RNA host gene 16 (SNHG16) [79], respectively (Figure 2E). Given that there are 1917 annotated hairpin precursors and 2654 mature sequences of human miRNAs according to the miRNA database, miRBase [80], it would not be a surprise to find intertwined networks of miRNA sponges sequestering miRNAs that target the 3′-UTR of the same gene for gene repression. It should be noted that miRBase has not been updated since October 2018 (Release 22.1); thus, more miRNAs have likely been identified since then, which may further contribute to more complex intertwined networks of miRNA sponges as more lncRNAs are discovered to function as such. Furthermore, it should be stressed that the interaction between lncRNAs and miRNAs stated in this subsection are restricted to the current findings (since 2019) in the liver. If the information search is extended to other tissues and time, more lncRNAs functioning as miRNA sponges with their specific targets can be identified. Taken together, the identification of lncRNAs as miRNA sponges is only at the beginning of understanding the many potential interactions among lncRNAs, miRNAs, and 3′-UTRs of protein-coding genes.

### 2.3. LncRNAs as Post-Transcriptional Modulators

Besides the above mechanisms of action, other biological roles for lncRNAs have also been reported. For example, the regulation of STAT3 is important in the liver as its activation is recorded in liver injuries and diseases [81,82,83]. Besides lncRNAs functioning as miRNA sponges to regulate the translation of *STAT3* (Figure 2E), the lncRNA Pvt1 oncogene (*PVT1*) increases the stability of phosphorylated STAT3 [84]; the lncRNA SLC2A1 antisense RNA 1 (*SLC2A1-AS1*) competitively binds to STAT3 to interfere its binding to the promoter of solute carrier family 2 member 1 (*SLC2A1*, also known as *GLUT1*) via forkhead box M1 (FOXM1) [85]; and the lncRNA TPTE pseudogene 1 (*TPTEP1*) inhibits the phosphorylation, homodimerization, and nuclear translocation of STAT3 [86]. Taken together, these recent findings about lncRNA mechanisms should facilitate the further understanding of pathophysiology of liver, which in turn, contributes to the development of new therapeutic approaches.

## 3. LncRNAs in HCC

Several lncRNAs have been implicated in the pathogenesis of HCC [87]. For example, the lncRNA metastasis-associated lung adenocarcinoma transcript 1 (*MALAT1*), which regulates alternative splicing, is upregulated in many solid tumors including HCC, and is associated with cancer metastasis and recurrence [88,89,90,91]. Furthermore, *MALAT1* was shown to promote cell metastasis by upregulation of transforming growth factor β-binding protein 3 (LTBP3) in HCC [92]. The *H19* gene is paternally imprinted and encodes an lncRNA that is expressed during development of fetal liver and strongly down-regulated after birth [93]. *H19* is highly expressed in many human cancers [94,95,96]; however, its function in HCC appears to be more complex compared to other cancers. Several studies have proposed that *H19* functions as an oncogene, with its activation contributing to the pathogenesis of HCC. For example, hypoxia has been shown to induce *H19* expression in HCC cells, whereas knockdown of *H19* attenuated HCC tumor growth [97]. On the other hand, *H19* is also a precursor of *miR-675*, and recent studies show that *miR-675* promotes tumor progression by repressing twist basic helix-loop-helix transcription factor 1 (*Twist1*), implying that the proposed oncogenic role of *H19* is mediated by the biological function of *miR-675* [98]. Interestingly, a study investigating the co-operation of *H19* and *miR-675* in HCC reported that knockdown of *H19* and *miR-675* promoted migration and invasion of HCC cells through the AKT/GSK-3β/Cdc25A signaling pathway [99], suggesting that *H19* and/or *miR-675* function as tumor suppressors.

The lncRNA taurine up-regulated 1 (*TUG1*) has been shown to regulate glycolysis and metastasis in HCC through hexokinase 2 (HK2), and thereby provides a potential therapeutic strategy of targeting TUG1 for treatment of HCC [100]. The *NEAT1* locus is highly conserved and produces two lncRNA isoforms, an abundant 3.7 kb *NEAT1_1* isoform and a 22.7 kb *NEAT1_2* isoform, which functions as a key structural component of nuclear paraspeckles, that are subnuclear bodies composed of RNA elements [101,102]. A recent study reported that *NEAT1* is a *bona fide* target gene of the tumor suppressor p53 and established a direct link among *NEAT1* paraspeckles, p53 biology, and tumorigenesis [103]. Notably, inhibition of *NEAT1_2* was shown to sensitize cancer cells to DNA-damaging chemotherapy, such as doxorubicin and bleomycin [103], whereas siRNA-mediated knockdown of *NEAT1* in HCC cell lines inhibited proliferation, migration, invasion, and induced apoptosis [104]. The expression of the lncRNA hepatocellular carcinoma up-regulated long non-coding RNA (*HULC*) is increased in HCC [105], and its expression is regulated by the transcription factor CREB (cAMP response element-binding protein) via interaction with *miR-372* [106]. *HULC* has been shown to contribute to HCC pathogenesis by deregulating lipid metabolism through a signaling pathway involving the peroxisome proliferator-activated receptor alpha (*PPARA*) [107]. In addition, *HULC* was reported to function as a miRNA sponge activating epithelial–mesenchymal transition, and promoting HCC progression and metastasis via the *miR-200a-3p*/ZEB1 signaling pathway [108].

The lncRNA differentiation antagonizing non-protein coding RNA (*DANCR*) is overexpressed in stem-like HCC cells, and binds to the 3′-UTR of β-catenin 1 (*CTNNB1*) mRNA, thereby blocking by competitive binding repression of *CTNNB1* by *miR-214*, *miR-320a*, and *miR-199a* [109]. This observation was confirmed in a mouse HCC model, suggesting that *DANCR* increases stemness features of HCC by derepressing *CTNNB1* [109]. The lncRNA HOX transcript antisense RNA (*HOTAIR*) is a part of the *HOXC* gene cluster on chromosome 12 and is an example of an lncRNA that functions as a scaffold and guides epigenetic regulators to genomic loci in *trans*. *HOTAIR* promotes silencing by acting as a scaffold to assemble the PRC2 and the lysine-specific demethylase 1 (LSD1) on the *HOXD* cluster, where these protein complexes specifically trimethylate histone H3 on lysine 27 and demethylate H3 on lysine 4, respectively, resulting in epigenetic silencing of *HOXD* genes [110,111]. *HOTAIR* is highly expressed in primary as well as metastatic breast tumors, and high level of expression in primary breast tumors is a powerful predictor of subsequent metastasis and death [112]. Several studies have investigated the clinical implications of *HOTAIR* in HCC. Expression of *HOTAIR* is significantly higher in HCC compared to adjacent normal liver tissues [113,114]. HCC patients that overexpress *HOTAIR* exhibit a shorter recurrence-free survival compared to patients with low levels of *HOTAIR* [115,116], suggesting that *HOTAIR* could be utilized as a metastatic biomarker for HCC.

## 4. LncRNAs in NAFLD and NASH

Accumulation of liver fat is a requisite for NAFLD and its progressive subtype NASH. A recent study showed that the lncRNA brown fat-enriched lncRNA 1 (*Blnc1*) is strongly upregulated in obesity and NAFLD in mice and functions as a core component of the LXR/SREBP1c pathway in the regulation of hepatic lipogenesis [117]. Furthermore, liver-specific ablation of *Blnc1* protected the *Blnc1* conditional knockout mice from high fat diet-induced hepatic steatosis and insulin resistance and ameliorated diet-induced NASH pathogenesis in mice, suggesting that *Blnc1* could be a potential therapeutic target for treatment of NAFLD and NASH [117]. Another recent study identified 89 differentially expressed lncRNAs in a NASH minipig model [118], whereas several lncRNAs, including the lncRNA functional intergenic repeating RNA element (*Firre*), have been shown to regulate adipogenesis [119]. *MALAT1* is frequently upregulated in HCC and is associated with cancer metastasis and recurrence [81,82,83,84], as described above. A recent study deployed a systems biology approach to explore lncRNA expression in the severity of NAFLD, and identified *MALAT1* as a potential molecular driver in the pathogenesis of NASH [120]. Another study profiled lncRNA expression in liver biopsies from NAFLD patients with normal liver histology, lobular inflammation, and advanced fibrosis, respectively, and found that the *HULC* and *MALAT1* lncRNAs were upregulated in fibrotic livers relative to normal liver tissue [121]. The C-X-C motif chemokine ligand 5 (CXCL5) was identified as a target for *MALAT1*, suggesting that *MALAT1* could play a role in the pathogenesis of NAFLD via a chemokine-mediated mechanism [121].

*MEG3* encodes an imprinted lncRNA that is frequently lost or downregulated in many human cancers and has been proposed to function as a tumor suppressor [122]. A systematic analysis of lncRNA expression in NAFLD and control liver samples found that the levels of *MEG3* were significantly decreased in liver tissues of NAFLD patients [123]. Downregulation of *MEG3* in NAFLD was negatively correlated with lipogenesis-related genes, while its overexpression in the human liver cancer cell line HepG2 reversed free fatty acid-induced lipid accumulation [51]. Mechanistic studies imply that *MEG3* functions as a miRNA sponge by competitively binding *miR-21*, leading to derepression of the miR-21 target LRP6, inhibition of the mTOR pathway, and subsequent induction of intracellular lipid accumulation [51]. These findings highlight the potential of *MEG3* as a biomarker and therapeutic target for NAFLD.

Expression profiling of lncRNAs of liver samples from high-fat diet fed mice identified 291 deregulated lncRNAs in a NAFLD mouse model compared to control animals [124]. Several lncRNAs, designated as fatty liver-related lncRNA (*FRL*) were associated with the PPAR signaling pathway via interactions with fatty acid binding protein 5 (Fabp5), lipoprotein lipase (Lpl) and fatty acid desaturase 2 (Fads2), respectively. The lncRNA fatty liver-related lncRNA 2 (*FLRL2*) is a nuclear-localized lncRNA that is down-regulated in the NAFLD mouse model, located in the intronic region of the aryl hydrocarbon receptor nuclear translocator-like (*Arntl*) gene and implicated in the regulation of *Arntl* in *cis* [124]. Overexpression of *FLRL2* in vitro alleviated lipid accumulation, inflammation, and ER stress in cultured mouse liver cells, whereas knockdown of *FLRL2* led to accumulation of cellular lipids [125]. Furthermore, adenovirus-mediated overexpression of *FLRL2* in high fat diet-fed mice resulted in activation of the Arntl-Sirtuin 1 pathway, inhibition of lipogenesis and reduction of hepatic steatosis, highlighting the therapeutic potential of *FLRL2* in NAFLD [125].

## 5. Therapeutic Targeting of lncRNAs for Treatment of Liver Disease

Several lncRNAs have been implicated in the pathogenesis HCC and NASH, as summarized above, which makes them potential targets for RNA-based therapeutics. Effective knockdown of disease-associated mRNAs can be achieved by using antisense oligonucleotides (ASOs) and siRNAs, and both approaches have also been deployed to target lncRNAs in functional studies in cell culture and *in vivo* [15,16]. However, the efficiency of ASO- and siRNA-based knockdown depends on the subcellular localization of the lncRNA [126]. For example, the nuclear-localized lncRNAs *MALAT1* and *NEAT1* were shown to be more effectively knocked down using ASOs, whereas the cytoplasmic lncRNAs *DANCR* and *OIP5-AS1* were more efficiently silenced using siRNAs [126]. Thus, it is important to determine the subcellular localization of a given lncRNA before embarking on functional studies using either ASOs or siRNAs.

ASOs are single-stranded oligonucleotides, typically 14–20 nucleotides in length, designed to bind and inhibit complementary RNA transcripts [127,128]. Most ASOs are chemically modified with phosphorothioate (PS) backbone linkages to enhance the pharmacokinetic properties of ASOs *in vivo* [127,128]. Moreover, improved stability and increased binding affinity are achieved using nucleotides with sugar modifications, such as 2’-O-methoxyethyl (MOE), 2’, 4’-constrained 2’-O-ethyl (cEt)-modified sugars, or locked nucleic acids (LNAs) [128]. Two different design paradigms are used for ASOs: gapmer ASOs for gene knockdown, in which a central deoxynucleotide region is flanked at both ends by modified nucleotides, e.g., 2′ MOE or LNAs, and uniformly modified mixmer ASOs that serve as steric blockers for e.g., miRNA inhibition or splice modulation [128]. Gene knockdown is mediated through RNase H activity, which binds and cleaves RNA/DNA heteroduplexes, whereas miRNA inhibition and splice modulation relies on steric blocking of the miRNA guide strand or specific splice sites, respectively, without causing RNA degradation [128]. Chemically modified ASOs have high potential as RNA-targeted therapeutics, which is highlighted by the FDA-approved ASO-based drugs, eteplirsen for treatment of Duchenne muscular dystrophy [129] and nusinersen for treatment of the neurodegenerative disease spinal muscular atrophy, respectively [130]. Furthermore, ASOs are frequently used to explore lncRNA function. For example, knockdown of *MALAT1* using intravenously injected cEt-modified gapmer ASOs in a mouse model of mammary carcinoma resulted in slower tumor growth and reduction in metastasis [17], whereas LNA-modified gapmer ASOs have been used to study the biological functions of the lncRNAs *NEAT1* and *SAMMSON*, respectively [103,131]. A recent study reported on systematic knockdown of 285 lncRNAs in human dermal fibroblasts deploying LNA gapmer ASOs followed by quantification of cellular growth, morphological changes, and transcriptomic responses using capped analysis of gene expression (CAGE) [132]. ASOs targeting the same lncRNAs showed global concordance, and the molecular phenotypes assessed by CAGE recapitulated the cellular phenotypes, while also providing information about the affected pathways [132].

Synthetic siRNA triggers can be used to engage the endogenous RNAi machinery to specifically knock down liver-expressed transcripts, including lncRNAs. As with ASOs, knockdown by siRNAs requires optimization for target specificity with minimized off-target effects and immunogenicity, as well as improved nuclease resistance. This can be achieved using different sugar modifications such as 2′-O-methyl and 2′-fluoro sugars, and by substituting phosphodiester linkages with PS linkages [133]. Furthermore, incorporation of a 5′(E)-vinyl-phosphonate into the guide strand enhances siRNA potency and duration of action *in vivo*, while the (S)-glycol nucleic acid modification in the seed region of the guide improves potency and decreases liver toxicity [133].

The liver is amenable to both non-targeted and targeted delivery of siRNAs, and has thus attracted a lot of interest from researchers and biopharmaceutical companies aiming to develop new therapeutic strategies for treatment of liver diseases. Non-targeted delivery utilizes siRNAs incorporated into lipid nanoparticles, which accumulate in the liver [133], whereas targeted delivery of siRNAs deploys the asialoglycoprotein receptor (ASGPR), which is expressed at high density on the surface of hepatocytes [134]. The GalNAc oligosaccharide has high affinity towards the ASGPR receptor, enabling rapid endocytosis and selective delivery of GalNAc-conjugated siRNAs to hepatocytes (reviewed in [135]). Indeed, the development of GalNAc-conjugated siRNA therapeutics has resulted in recent FDA approval of the siRNA-based drug, givosiran, for treatment of acute hepatic porphyria [136]. Furthermore, targeted delivery of triantennary GalNAc-conjugated gapmer ASOs was shown to improve ASO potency 10-fold in mice [137], which, together with the recent advances in developing GalNAc-conjugated siRNA drugs, sets the stage for discovering novel lncRNA-targeted therapeutics for HCC and NASH.

## 6. Concluding Remarks

This review summarizes the recent findings of lncRNAs in HCC, NASH, and NAFLD. By introducing different mode of actions of lncRNAs in the liver, we particularly focused on the lncRNAs whose dysregulation leads to the pathogenesis of HCC and NASH. Although lncRNAs are the focus of this review, other classes of ncRNAs apart from miRNAs and lncRNAs have also been reported in liver disease. For example, the small ncRNA, small Cajal body-specific RNA 10 (*SCARNA10*), binds to PRC2 to positively regulate TGFβ signaling in hepatic fibrogenesis [138]. Other small RNAs and their lncRNA host genes are also reported, such as: small nucleolar RNA host gene 1 (*SNHG1*), which binds to DNMT1 to inhibit p53 expression [139]; and its family member, small nucleolar RNA host gene 20 (*SNHG20*), which interacts with PTEN to negatively regulate its protein level [140]. As the number of lncRNAs is higher than that of protein-coding genes [141], more intensive, functional studies of lncRNAs, especially using genetic manipulation (e.g., CRISPR/Cas9-based knockouts, knockout and transgenic mice) and knockdown by ASOs or siRNAs, are needed to confirm previous findings on lncRNAs in the liver aimed at understanding their biological mechanisms and to guide new therapeutic approaches to combat life-threatening liver diseases, such as HCC and NASH.

## Figures and Tables

**Figure 1 ncrna-06-00034-f001:**
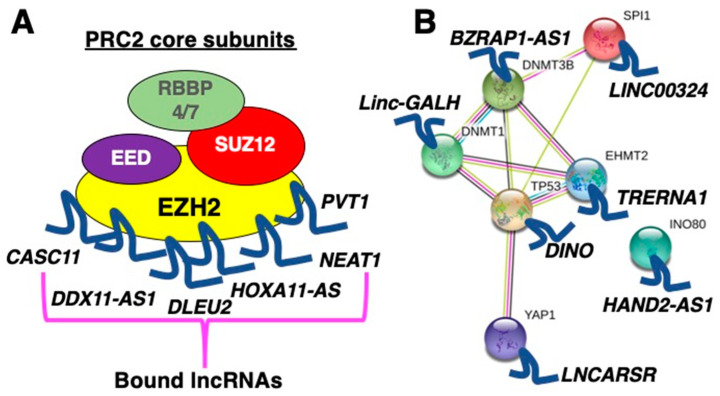
Molecular mechanism of long non-coding RNAs (lncRNAs). (**A**) A growing list of lncRNAs bind to the enhancer of zeste homolog 2 (EZH2). Because of promiscuous RNA binding capability of EZH2, a large number of lncRNAs have been identified as its binding partner. EED, embryonic ectoderm development; RBBP4, RB binding protein 4, chromatin remodeling factor; RBBP7, RB binding protein 7, chromatin remodeling factor; SUZ12, SUZ12 polycomb repressive complex 2 subunit. (**B**) LncRNAs binding to epigenetic and transcription factors. The protein–protein interactions (PPI) are based on the information provided by the STRING database [38].

**Figure 2 ncrna-06-00034-f002:**
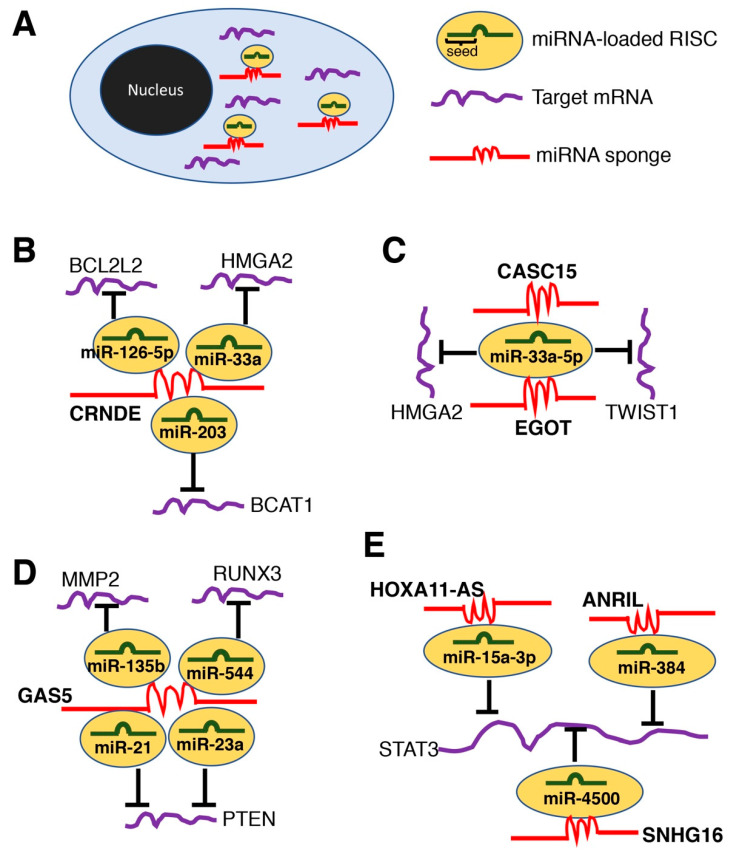
LncRNAs as miRNA sponges. (**A**) A schematic illustration of miRNA sponge mechanism in the cell. RISC, RNA-induced silencing complex. (**B**) The lncRNA colorectal neoplasia differentially expressed (*CRNDE)*. (**C**) An example of miRNA being sequestered by two lncRNAs. *CASC15*, cancer susceptibility 15; *EGOT*, eosinophil granule ontogeny transcript; *HMGA2*, high mobility group AT-hook 2; *TWIST1*, twist family bHLH transcription factor 1. (**D**) The lncRNA growth arrest specific 5 (*GAS5*). Interestingly, GAS5 sequesters *miR-21* and *miR-23a*, which both target phosphatase and tensin homolog (*PTEN*) mRNA. *BCL2L2*, BCL2 like 2; *BCAT1*, branched chain amino acid transaminase 1; *MMP2*, matrix metallopeptidase 2; *RUNX3*, RUNX family transcription factor 3. (**E**) An example of a target mRNA with multiple miRNA binding sites. *HOXA11-AS*, HOXA11 antisense RNA; *ANRIL*, antisense non-coding RNA in the INK4 locus; *SNHG16*, small nucleolar RNA host gene 16; *STAT3*, signal transducer and activator of transcription 3.

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
