# Peer review of "Long Non-Coding RNAs in Liver Cancer and Nonalcoholic Steatohepatitis"

_ncrna, 2020, doi:10.3390/ncrna6030034_

Round 1

Reviewer 1 Report

Uchida and Kauppinen

Long non-coding RNAs in liver cancer and nonalcoholic steatohepatitis

In this review article, the authors have summarized the recent literature on non-coding RNA molecules involved in liver diseases such as HCC, NASH and NAFLD. The first half of the manuscript is dedicated to general aspects of liver disease and the molecular functions of lncRNAs. The second half highlights lncRNAs in HCC, in NASH and in NAFLD. The review ends with a concluding remarks chapter that mentions several other classes of non-coding RNAs that have not been discussed in the manuscript.

This is a timely and comprehensive short review on lncRNAs in several liver diseases. LncRNAs are not understood and of high interest regarding disease understanding and treatment. However, there are a number of weaker points, which are summarized below.

  1. Overall, the manuscript presents a large number of publications, which are simply added one by one. Many of these papers are published in highly specialized small journals without any further confirmation by other labs. It would be desirable that such papers are discussed more critically or a panel of ‘solid and validated’ publications was selected and discussed in more detail. Especially the concept of miRNA sponges is debated and there are many papers claiming such effects without convincing data. For example, conclusions solely drawn from relative qPCR measurements would require further work.

  1. It is also not fully clear what one should conclude from this review. There are many lncRNAs involved in the liver diseases, this becomes clear. But there is not much offered that would go beyond this fact. Maybe the authors could add a short chapter on potential strategies of targeting such RNAs and discuss the current stage of drug development? It could also be part of the last concluding remarks chapter.

  1. Some parts are a little redundant. Meg3 as sponge, for example, is discussed in the miRNA sponge chapter and later on also in the NAFLD and NASH chapter. It is clear that it is presented in different context but it appears nevertheless redundant.

  1. Concluding remarks chapter, first sentence: “This review summarizes…in liver disease”. This should be more specific. Only HCC, NASH and NAFLD are included and there might be additional liver diseases.

  1. Concluding remarks chapter: small nucleolar RNA host genes are not small RNAs. If at all functional, these host genes are rather lncRNAs containing a snoRNA (which is also not classified as small RNA). This should be corrected.

  1. Concluding remarks chapter, lane 318: there is a typo that should be corrected.

Author Response

In this review article, the authors have summarized the recent literature on non-coding RNA molecules involved in liver diseases such as HCC, NASH and NAFLD. The first half of the manuscript is dedicated to general aspects of liver disease and the molecular functions of lncRNAs. The second half highlights lncRNAs in HCC, in NASH and in NAFLD. The review ends with a concluding remarks chapter that mentions several other classes of non-coding RNAs that have not been discussed in the manuscript.

This is a timely and comprehensive short review on lncRNAs in several liver diseases. LncRNAs are not understood and of high interest regarding disease understanding and treatment. However, there are a number of weaker points, which are summarized below.

Overall, the manuscript presents a large number of publications, which are simply added one by one. Many of these papers are published in highly specialized small journals without any further confirmation by other labs. It would be desirable that such papers are discussed more critically or a panel of ‘solid and validated’ publications was selected and discussed in more detail. Especially the concept of miRNA sponges is debated and there are many papers claiming such effects without convincing data. For example, conclusions solely drawn from relative qPCR measurements would require further work.

Response: Compared to other fields (e.g., cardiovascular, hematology, and neuroscience), the knowledge of lncRNAs in the liver and liver diseases is very limited with only very few functional and mechanistic studies. Thus, the main objective of this review is to provide an overview of the current status of lncRNAs in the liver and liver diseases, especially focusing on HCC and NAFLD/NASH. With regard to the latter point about the controversies regarding the miRNA sponge/ceRNA hypothesis, we have added the following paragraph to section 2.2.:

“It should be noted that the ceRNA hypothesis is still being debated due to the fact that most experimental evidence is based on expression profiling such as qRT-PCR. Furthermore, studies that have modelled transcriptome-wide miRNA target site abundance suggest that physiological changes in expression levels of most individual transcripts, including lncRNAs, are insufficient to modulate miRNA activity [63-66]. Thus, further research is needed to establish the miRNA sponge (ceRNA) mechanism for lncRNAs.”

It is also not fully clear what one should conclude from this review. There are many lncRNAs involved in the liver diseases, this becomes clear. But there is not much offered that would go beyond this fact. Maybe the authors could add a short chapter on potential strategies of targeting such RNAs and discuss the current stage of drug development? It could also be part of the last concluding remarks chapter.

Response: We agree with the reviewer’s comments above; thus, we have added a new section (Section 5) about therapeutic targeting of lncRNAs using antisense oligonucleotides and siRNAs.

Some parts are a little redundant. Meg3 as sponge, for example, is discussed in the miRNA sponge chapter and later on also in the NAFLD and NASH chapter. It is clear that it is presented in different context but it appears nevertheless redundant.

Response: We agree with the reviewer. However, MEG3 is one of the first lncRNAs that were characterized and is widely studied in different areas, including in the liver. Thus, MEG3 has been mentioned in two different subsections in this review to highlight its importance.

Concluding remarks chapter, first sentence: “This review summarizes…in liver disease”. This should be more specific. Only HCC, NASH and NAFLD are included and there might be additional liver diseases.

Response: The above sentence is corrected as below:

“This review summarizes the recent findings of lncRNAs in HCC, NASH, and NAFLD.”

Concluding remarks chapter: small nucleolar RNA host genes are not small RNAs. If at all functional, these host genes are rather lncRNAs containing a snoRNA (which is also not classified as small RNA). This should be corrected.

Response: The above point is corrected as below:

“Other small RNAs and their lncRNA host genes are also reported, such as: small nucleolar RNA host gene 1 (SNHG1), which binds to DNMT1 to inhibit p53 expression [123]; and its family member, small nucleolar RNA host gene 20 (SNHG20), which interacts with PTEN to negatively regulate its protein level [124].”

Concluding remarks chapter, lane 318: there is a typo that should be corrected.

Response: The above typo has been corrected.

Reviewer 2 Report

This is very well written review and figures are very interesting. The authors are to be commended for this great work. Congratulations!

The only major comment, according to a recent international consensus the nomenclature of NAFLD has been updated to metabolic associated fatty liver disease (MAFLD). Please, update the title and entire manuscript accordingly and refer to the reference. NASH should be written as steatohepatitis (PMID: 32278004 and PMID: 32044314).

Author Response

This is very well written review and figures are very interesting. The authors are to be commended for this great work. Congratulations!

The only major comment, according to a recent international consensus the nomenclature of NAFLD has been updated to metabolic associated fatty liver disease (MAFLD). Please, update the title and entire manuscript accordingly and refer to the reference. NASH should be written as steatohepatitis (PMID: 32278004 and PMID: 32044314).

Response: We would like to thank the reviewer for the positive feedback. Regarding the issue with nomenclatures, you are absolutely right about the new nomenclatures. However, the old terminologies are still used widely in the literatures. To highlight this fact, we modified the following sentence in the Introduction section with the suggested citations:

“Major risk factors for HCC include chronic hepatitis B virus (HBV) and hepatitis C virus (HCV) infections [1,4], alcohol-induced liver disease, non-alcoholic fatty liver disease (NAFLD), which was recently proposed to be renamed as metabolic associated fatty liver disease (MAFLD) [5] and non-alcoholic steatohepatitis (NASH) [6,7]”